# Population Pharmacokinetics of Colistin Methanesulfonate Sodium and Colistin in Critically Ill Patients: A Systematic Review

**DOI:** 10.3390/ph14090903

**Published:** 2021-09-06

**Authors:** Mohd Shafie Zabidi, Ruzilawati Abu Bakar, Nurfadhlina Musa, Suzana Mustafa, Wan Nazirah Wan Yusuf

**Affiliations:** 1Department of Pharmacology, Health Campus, School of Medical Sciences, Universiti Sains Malaysia, Kubang Kerian, Kelantan 16150, Malaysia; mszpharm@gmail.com (M.S.Z.); ruzila@usm.my (R.A.B.); 2Human Genom Centre, Health Campus, School of Medical Sciences, Universiti Sains Malaysia, Kubang Kerian, Kelantan 16150, Malaysia; fadhlina@usm.my; 3Department of Pharmacy, Hospital Raja Perempuan Zainab II, Kota Bharu, Kelantan 16150, Malaysia; szn_m@yahoo.com

**Keywords:** colistin methanesulfonate sodium, colistin, critically ill patients, population pharmacokinetics, therapeutic drug monitoring

## Abstract

Understanding the pharmacokinetics parameter of colistin methanesulfonate sodium (CMS) and colistin is needed to optimize the dosage regimen in critically ill patients. However, there is a scarcity of pharmacokinetics parameters in this population. This review provides a comprehensive understanding of CMS and colistin pharmacokinetics parameters in this population. The relevant studies published in English that reported on the pharmacokinetics of CMS and colistin from 2000 until 2020 were systematically searched using the PubMed and Scopus electronic databases. Reference lists of articles were reviewed to identify additional studies. A total of 252 citation titles were identified, of which 101 potentially relevant abstracts were screened, and 25 full-text articles were selected for detailed analysis. Of those, 15 studies were included for the review. This review has demonstrated vast inter-study discrepancies in colistin plasma concentration and the pharmacokinetics parameter estimates. The discrepancies might be due to complex pathophysiological changes in the population studied, differences in CMS brand used, methodology, and study protocol. Application of loading dose of CMS and an additional dose of CMS after dialysis session was recommended by some studies. In view of inter-patient and intra-patient variability in colistin plasma concentration and pharmacokinetics parameters, personalized colistin dosing for this population is recommended.

## 1. Introduction

Colistin is an antibiotic that is used as a last resort to treat infections caused by multidrug-resistant Gram-negative bacteria. Colistin is given intravenously as the inactive prodrug colistin methanesulfonate sodium (CMS), which is transformed to a variety of partially sulfomethylated derivatives in the body before being converted to colistin [1]. Colistin’s pharmacokinetics and clinical application have been studied extensively in the past. Unfortunately, studies have found inter- and intra-individual variability in the pharmacokinetics of colistin, resulting in extremely varied plasma concentrations following the same dosage schedule [2,3,4,5]. In vivo, CMS has complex pharmacokinetics and variable bioconversion, particularly in patients with varying degrees of renal function [2,6]. CMS conversion to colistin is slow, while CMS excretion from the body system is high in individuals with good renal function, making it difficult to achieve the desired colistin plasma concentration (2–4 mg/L) [7]. For polymyxin-induced bacterial death, rapid achievement of the desired concentration has been demonstrated to be crucial [8].Earlier pharmacokinetic results revealed that starting colistin therapy with a loading dosage resulted in faster target concentration achievement and better clinical outcomes [3,9,10]. However, other studies have questioned the need for a loading dose [11,12]. Despite receiving a suitable loading dosage, critically ill patients with good renal function had trouble obtaining the desired concentration [7]. Furthermore, because colistin is extremely nephrotoxic and acute kidney damage (AKI) occurs frequently with conventional doses, starting colistin therapy with a higher dose might compromise the patients’ safety [13].

More investigations on the pharmacokinetics of CMS and colistin in critically ill patients, as well as their relationship to clinical efficacy and renal function, have recently been published [2,4,5]. Clinical failure has been linked to subtherapeutic colistin concentrations [5], although high colistin exposure has not been linked to longer survival [2]. Many studies have linked a high colistin concentration to acute renal damage [2,4]. Understanding how variables influenced CMS and colistin pharmacokinetic variability should help to improve therapy and avoid toxicity. The safety outcome was substantially associated with pharmacokinetic measures, including maximum colistin concentration, the area under the plasma concentration curve for 8 h, apparent total body clearance, and apparent volume of distribution [4]. Many research used pharmacometrics methods to assess the effects of variable factors on CMS and colistin distribution [2,3,14]. Despite the fact that several of the factors were not clinically significant, their combined impact must be considered. As a result, these factors must be investigated further.

Since colistin concentrations in epithelial lining fluid (ELF) were too low or undetectable following intravenous administration of CMS compared to aerosol delivery, pneumonia, the most prevalent hospital-acquired infection in ICU, proved to be the largest difficulty in colistin therapy [15]. As a result, patients with hospital-acquired pneumonia (HAP) or ventilator-acquired pneumonia (VAP) should get additional colistin aerosol therapy [16]. In a recent study of critically ill patients receiving intravenous colistin with a loading dose and adjunctive colistin aerosol, the combination of intravenous colistin with a loading dose and adjunctive colistin aerosol was linked to improved 30-day mortality and microbiological outcome without increased nephrotoxicity in critically ill patients with HAP and VAP [17]. In this cohort, it was advised that an intravenous and adjunctive colistin aerosol combination be examined as a therapeutic option [17].

Despite its toxicity, colistin is being utilized to treat critically ill individuals, according to recent data. Understanding the pharmacokinetic parameters of CMS and colistin variability requires a scientific analysis of the available data. To ensure the optimal dosage of CMS is utilized in patient management for greater efficacy and to avoid toxicity, an understanding of the variability in the pharmacokinetic characteristics of CMS and colistin is essential. To our knowledge, no systematic review or meta-analysis of CMS and colistin population pharmacokinetics in critically ill patients has been published. Therefore, we conducted a systematic evaluation of published data characterizing CMS and colistin pharmacokinetics parameters in critically ill patients. The purpose of this research is to review and discuss the existing data in order to gain a thorough grasp of the CMS and colistin pharmacokinetics characteristics. Our findings will help critical care professionals optimize colistin dosing regimens in critically ill patients while avoiding toxicity.

## 2. Materials and Methods

### 2.1. Search Strategy for Identification of Studies

Studies were identified in accordance with The Preferred Reporting Items for Systematic Reviews and Meta-Analyses (PRISMA) 2020 statement [18]. A comprehensive search of relevant studies published from 2000 to 2020 that reported on the pharmacokinetics of CMS and colistin after intravenous administration of CMS was conducted using the PubMed and Scopus electronic databases. In addition, the reference lists of narrative reviews and selected articles were hand-searched for relevant studies. For the database searches, titles and abstracts were searched using the terms “critically ill” OR “critical illness” OR “critical care” OR “intensive care” OR “intensive care unit” OR “ICU” AND “colistin” OR “colistin methanesulfonate sodium” OR colistimethate sodium” AND “pharmacokinetics.” The search was limited to human studies and articles published in the English language.

#### 2.1.1. Study Selection and Study Eligibility

The scope of the systematic literature review was based on population, intervention, outcome, and study design, as summarized in Table 1. One author independently screened the titles and/or abstracts of the studies against the scope and inclusion criteria. During the screening, the number of excluded articles was documented. The study identification and selection are illustrated in Figure 1. When a definite decision could not be made based on the title and/or abstract alone, the full paper was obtained for detailed assessment against inclusion criteria. Potentially eligible studies were identified by consensus between three authors, and when discrepancies could not be resolved, another two authors were consulted. The full text of potentially eligible studies was retrieved and assessed for eligibility. Reference management was performed in Mendeley Desktop version 1.19.4. (Elsevier Inc., New York, NY, USA).

#### 2.1.2. Inclusion and Exclusion Criteria

Population pharmacokinetics studies and/or prospective clinical studies were included if the pharmacokinetics data were collected from adult patients receiving intravenously administered CMS. Studies with concomitant inhalation administration were also included. However, pharmacokinetic data other than in plasma or from other than intravenous administration was not reported. The articles clearly described the study population, CMS dosing regimen, bioanalytical methods, and statistical tools used that were included in the study. The only pharmacokinetics studies included were ones that were conducted with liquid chromatography coupled with mass spectrometry or fluorimetry detection assays which are able to quantify CMS and colistin separately. Studies that collected data relating to pediatric populations and non-human studies were excluded. Studies related to the subpopulation of critically ill patients (e.g., burn injury patients, head injury patients, etc.) that required specialized units were excluded since the pharmacokinetics parameters in this subpopulation were different from those in other critically ill patient groups.

### 2.2. Data Extraction and Quality Assessment

A data extraction table was designed, and three reviewers independently extracted data from the included studies. Any discrepancies observed between the data extracted by the three authors were resolved through discussion, and when the discrepancies could not be resolved, the fourth and fifth authors were consulted. Data management was performed using Microsoft Excel version 2019. For all the included studies, data extracted from the literature include:Study characteristics: authors, year, study size, study population, drug manufacturer/brand, CMS preparation, and bioanalytical method;Patient characteristics: sample collection, age, body weight, creatinine clearance (CrCl), and number of renal impaired patients;Outcomes: program for pharmacokinetic analysis, pharmacokinetic modeling, CMS dose, maximum concentration (C_max_), steady-state concentration (C_ss_), half-life (t_1/2_), clearance (Cl), and volume of distribution (V_d_).

So far, no validated tool is available to assess the methodological quality in pharmacokinetics studies. The quality of the studies, therefore, was assessed based on the papers that were published in peer-reviewed journals. In addition, studies with a very small sample size (less than 10 patients) were not selected.

### 2.3. Data Synthesis and Analysis

Demographic data in the selected published papers were presented as means and standard deviations (SD). For data that was presented as median (IQR) and the raw data were not available, mean (SD) were estimated using a method described by Wan et al. [19]. In studies where the data was presented from pharmacokinetics modeling, the summary value of pharmacokinetics data or predicted value from the modeling were obtained for estimation and comparison. All relevant data were summarized descriptively.

## 3. Results

The literature search from electronic databases and hand-searches yields 252 citation titles, of which 101 potentially relevant abstracts were screened, and 25 full-text articles were selected for detailed analysis. Of those, 15 studies were included for the review (Appendix A, available as Appendix A).

### 3.1. Demographic Study and Patient Characteristics

A total of 707 patients were enrolled in the pharmacokinetics studies that were conducted from different study sites, such as Greece [3,9,10,20,21], France [11,15,22], Italy [23], Switzerland [14], Israel [2], India [4,5,12], United States of America [6], and Thailand [6]. Most of the studies involved patients with preserved renal function; only one study evaluated patients with various renal function groups [6], and three studies evaluated patients who received renal replacement therapy [14,21,22]. Most of the pharmacokinetics studies of critically ill patients involved small sizes, except studies from Kristoffersson et al. [2], Grégoire et al. [11], and Garonzik et al. [6], who used larger sample sizes. The mean age was between 50–65 years old, and three studies involved a mean age of less than 50 years old [4,12,23]. The reported mean creatinine clearance (CrCl) value between studies was 80–120 mL/min. A total of 188 patients had CrCl less than 50 mL/min at the baseline [2,3,6,9,10,12,14,20,21,22] and of those, 136 patients required renal replacement therapy [2,6,14,21,22].

In addition to intravenous CMS, four studies also administered inhalation CMS [4,6,15,22]. Two studies used a CMS dosage that was determined based on body weight [12,14], two studies were based on CrCl [2,3], four studies were freely chosen by the physician [4,5,8,10], and seven studies used the same dose on all patients [4,5,10,15,20,21,23]. A different type of CMS brand was used, and most of the CMS preparation prior to the administration was less than 80,000 IU/mL, except in the Karaiskos et al. study [3]. Eleven studies evaluated the pharmacokinetic parameter of CMS and colistin [2,5,6,9,11,15]. Four studies examined the pharmacokinetic parameter of colistin in plasma after an intravenous administration of CMS; however, CMS pharmacokinetics parameters were not evaluated [4,12,20,23]. A total of 25 pharmacokinetics studies were identified, with nine studies collecting a blood sample after the first dose and after repeated doses of CMS [2,5,9,12,21]. Of those, five studies used the first dose as a loading dose [2,3,5,10]. Most of the studies reported pharmacokinetics data as mean or median, and nine studies reported pharmacokinetics parameters derived from modeling analyses [2,3,6,9,10,11,14,15,22].

### 3.2. Concentration of CMS and Colistin in Plasma after CMS Administration

The reported CMS and colistin concentrations in critically ill patients’ plasma after a first dose with and without a loading dose are summarized in Table 2. A total of nine studies reported a maximum concentration (C_max_) value of CMS and/or colistin after single-dose intravenous administration of CMS [2,3,5,9,10,11,12,21,22]. Of the reported articles, five studies reported C_max_ values of CMS and/or colistin after application of a loading dose [2,3,5,10,22].

#### 3.2.1. Concentration of CMS and Colistin in Plasma after the First Dose of CMS Administration without a Loading Dose

In the single-dose study by Karvanen et al. [21], patients receiving 2 MIU of CMS reported CMS C_max_ value of 6.9 mg/L. However, Plachouras et al. [9], despite using a higher dose (3 MIU) of CMS with the same brand, reported lower C_max_ values (3.5 mg/L). The difference between both studies was the patients recruited in the Karvanen et al. [21] study presented with impaired renal functions, whereas those in the study by Plachouras et al. [9] had a normal renal function. Thus, the possible reason for the high CMS concentration in the study by Karvanen et al. [21] was renal impairment. Colistin methanesulfonate sodium is either directly excreted in urine or converted systemically into colistin. Any reduction in renal function results in reduced renal clearance of CMS and a greater proportion of CMS available in the body [6] to be converted to colistin; hence, more colistin is formed. In preserved renal functions, the CMS concentration declined with time in a mono-exponential or bi-exponential manner, depending on the study [9,11], leading to lower colistin levels.

Colistin methanesulfonate sodium undergoes hydrolysis in vivo to form colistin. Colistin concentrations in plasma increased slowly with time. The time to reach maximal plasma concentrations (T_max_) and C_max_ varied between reported studies [4,5,10,20]. Plachouras et al. [9] and Karvanen et al. [21] found that suboptimal plasma concentration (C_max_ < 2 mg/L) was achieved 6–7 h after infusion. However, Grégoire et al. [11] reported C_max_ of 2 mg/L at 3 h after the end of infusion, and Karnik et al. [12] observed high plasma concentration (C_max_ of 8.7 ± 6 mg/L) at 1 h after the end of infusion. As a consequence of the slow CMS conversion observed, patients in Plachouras et al. [9], were exposed to suboptimal plasma colistin concentration for 2–3 days before reaching a steady-state due to prolonged half-life (14.4 h). Therefore, Plachouras et al. [9] suggested the need for a loading dose and a change in the dosing strategy for CMS. Plachouras et al. [9] predicted that a CMS loading dose of 9–12 MIU would achieve the targeted concentration faster. However, faster CMS conversion was observed in two other pharmacokinetic studies: Grégoire et al. [11] and Karnik et al. [12].

The discrepancies in the rate of concentration attainment of colistin were due to the variety of CMS formulations between different CMS brands used in the studies [24]. Studies that used CMS with brand name Colistin (Norma, Greece) reported that T_max_ was between 6–8 h after infusion to reach C_max_ [3,9,10,21]. In other studies that used CMS with brand Colimycine (Sanofi-Aventis), the reported T_max_ was between 3–6 h after infusion to reach C_max_ [11,22], but Kristofferson et al. [2] observed at 45 min after the start of the loading dose infusion the measured colistin concentration was low (Table 2). In a single-dose study, Karnik et al. [12] discovered a high C_max_ within 1 h after infusion. The mean C_max_ in Karnik et al. [12] was high due to the wide range of colistin C_max_ reported in the study (median 4.6 (2.5–23.2) mg/L). The use of different CMS formulations may have contributed to the interstudy discrepancies. However, further study is required to support this view as Karaiskos et al. [3] observed that CMS slowly converted over time for all six brands, including the brand used by Gregoire et al. [11].

The amount of CMS to use and the time it takes to reach the required steady-state colistin concentration have been a point of contention in the past [3,9,11]. A loading dose is required for a CMS brand that undergoes slow conversion; however, it is impossible to determine the rate of in vivo conversion for a particular brand [16]. Therefore, the therapeutic benefits of a loading dose may outweigh the potential risk of acute kidney injury (AKI) associated with a loading dose [16].

#### 3.2.2. Concentration of Colistin in Plasma after Application of a Loading Dose

The study by Mohamed et al. reported that after 6 MIU of CMS loading dose given to critically ill patients, the concentrations were on average 1.34 mg/L (range, 0.374 to 2.59 mg/L) at 8 h [10]. The study also concluded that although colistin concentration improved with CMS loading dose of 6 MIU, bacterial killing required a higher loading dose (6–9 MIU) to kill the wild-type *Pseudomonas aeruginosa* strain with a minimum inhibitory concentration (MIC) of 1 mg/L achieved 1 h earlier (e.g., at 5.5 h) [10].

A subsequent study by Karaiskos et al. [3] was the first to demonstrate colistin concentrations above the MIC breakpoints after administration of 9 MIU CMS loading dose followed by a randomized infusion time of either 0.5 or 1 h, where the C_max_ was above 2 mg/L within the first hours of the initiation. The colistin concentration in observed range from 0.95 to 5.1 mg/L and 0.68 to 8.72 mg/L after the loading dose of 9 MIU and at a steady-state, respectively [3]. Both of the studies showed inter-individual variability (IIV) in colistin disposition [3,10]. Mohamed et al. [10] estimated the IIV as 76 percent, and Karaiskos et al. [3] estimated it as 71 percent. The variability in concentration between dosing administration (IOV) has also been determined in critically ill patients from previous studies [6,10]; therefore, the suggestion of TDM [25] for individual dosing may have no benefit [3].

Moni et al. [5] studied the safe and effective use of colistin with an application of loading dose. In their study, “clinical cure group” was defined as patients who resolved from signs and symptoms of infection at the end of colistin therapy, and “clinical failure group” were those with persistence or worsening of signs and symptoms of infection. It was reported that the C_max_ colistin after loading dose was 3 ± 1.1 mg/L for the “clinical cure group” and 2.37 ± 1.2 mg/L for the “clinical failure group” (*p* = 0.13), while the mean steady-state concentration (C_ssave_) was 2.25 ± 1.3 mg/L and 1.78 ± 1.1 mg/L in “clinical cure group” and “clinical failure groups”, respectively (*p* = 0.19) [5]. For both groups, the results demonstrated that the desired colistin concentration was obtained when the loading dose was applied. However, in the “clinical failure group”, the steady-state concentration was found to be subtherapeutic. When C_ssave_ levels exceeded 2 mg/L, no substantial renal toxicity was seen, according to the study. The study emphasizes the importance of using TDM to guide colistin dosing.

Kristoffersson et al. [2] reported that the population pharmacokinetic model predicted that only 66 percent of the patients had colistin concentrations of more than 2 mg/L at 4 h after starting treatment with application of loading dose [2]. The observed colistin concentration was more than 2 mg/L in the majority of patients with CrCl less than 120 mL/min, while a higher dose was needed to achieve the same exposure in patients with CrCl of more than 120 mL/min [2]. The study concluded that high colistin exposure was associated with poor kidney function and was not related to prolonged survival [2].

The application of a loading dose of 9 MIU CMS made it possible to achieve the targeted colistin concentration in critically ill patients but may not necessarily improve the survival outcome [2,7,16,26]. Since colistin had a narrow therapeutic index and wide inter-individual variability, individualized colistin dosage guided by colistin exposure in the blood may be helpful to balance between therapeutic efficacy and the occurrence of toxicity. In addition, International Consensus Guidelines for the Optimal Use of the Polymyxins [16] recommended a study of the exposure-response relationship between colistin in plasma and antibacterial effects [27], the risk of AKI [28,29], TDM, and AFC for the management of critically ill patients on colistin.

#### 3.2.3. Concentration of Colistin in Plasma after Repeated Doses

The CMS dose regimen given between studies ranged between 1–3 MIU at 8–12 h. The average colistin concentration at a steady state is determined by its rate of formation and its rates of elimination. Colistin disposition depends on the fraction of CMS dose (f_m_) that converted to colistin. Plachouras et al. [9] observed slow conversion of CMS to colistin (7–8 h); however, other two pharmacokinetics studies [11,12] reported more rapid conversion (within one hour). Therefore, there were discrepancies in the times to achieve a steady-state concentration of colistin in critically ill patients [9,11]. Regarding the steady-state concentration achieved, Plachouras et al. [9] reported a longer time (2–3 days) compared to Grégoire et al. [11], which was achieved within a day.

Garonzik et al. [6] reported the steady-state concentration (C_ssave)_ of colistin in plasma in critically ill patients ranged from 0.48 to 9.38 mg/L. The pharmacokinetic model Garonzik et al. [6] described showed only 35 percent of cases achieved the targeted concentration at a steady state. There was a wide variation of C_ssave_ with values ranging from 0.68 to 8.72 mg/L after applying 9 MIU CMS and followed with 4.5 MIU every 12 h as a maintenance dose, as explained by Karaiskos et al. [3]. Therefore, only 33 percent of the patients achieved colistin above 2 mg/L [3]. There is a large variability of C_max_ colistin values between patients due to individual differences in CrCl in the study. In Karaiskos et al. [3], the maintenance dose commenced 24 h after the administration of the CMS loading dose as Garonzik et al. [6] proposed. As a result, only one-third of the study patients reached the target concentration [3].

Pharmacokinetic/pharmacodynamic (PK/PD) analysis suggests that it is possible the colistin concentration declined below the desired concentration by 12 h [30]. Therefore, the dosing interval of 8–12 h between loading dose and maintenance was suggested. In a recent pharmacokinetics study by Kristofferson et al. [2], with a large number of patients (*n* = 349) and patients receiving a loading dose of 9 MIU followed by 4.5 MIU every 12 h, more than 90 percent of the patients had colistin concentration above 2 mg/L at steady state. Unfortunately, the study did not mention the actual time they started the maintenance dose after initiating the loading dose.

#### 3.2.4. Concentration of Colistin in Patients with Renal Impairment and Patients Receiving Renal Replacement Therapy

Garonzik et al. [6] studied the plasma colistin concentration in 105 critically ill patients in various categories and developed the first scientifically-based dosing strategies for CMS to achieve a targeted C_ssave_. The average C_ssave_ was 2.36 mg/L (range 0.48–9.38 mg/L) in all critically ill patients. When the maintenance dosing suggestions for various categories of critically ill patients were applied to all patients using the colistin C_ssave_ target of 2.5 mg/L, among those not on any renal replacement plus those on HD, 3/101 patients are predicted to have achieved a colistin C_ssave_ of 0.5 to 1 mg/L, 86/101 will have a colistin C_ssave_ 1 to 4 mg/L, and 12/101 will have a colistin C_ssave_ of more than 4 mg/L. All 12 of the patients with HD would be predicted to achieve a C_ssave_ between 1.9 to 3.4 mg/L. All four CRRT patients would be predicted to achieve a C_ssave_ concentration of colistin between 1.9 to 4.2 mg/L.

The population pharmacokinetics study of colistin and relation to survival in critically ill patients by Kristoffersson et al. [2] reported that concentration of colistin after a repeated dose was more than 2 mg/L in 94 percent of patients with CrCl of less than 120 mL/min, and 44 percent of patients with CrCl more than 120 mL/min. Patients with CrCl less than 50 mL/min who received an adjusted maintenance dose had similar concentrations as patients with CrCl of 50–80 mL/min. Ninety-five percent of patients with CrCl 50–79 mL/min, 83 percent of patients with CrCl 80–119 mL/min, and 44 percent of patients with CrCl more than 120 mL/min had a measured colistin concentration of more than 2 mg/L. The study found that colistin concentrations determined after 10 h after a maintenance dose was lower in patients with higher CrCl values. Fifty-eight percent of patients with CrCl 50–79 mL/min, 37 percent of patients with CrCl 80–119 mL/min, and 11 percent of patients with CrCl more than 120 mL/min had a measured colistin concentration of more than 4 mg/L. The studies concluded that high colistin exposure was associated with poor kidney function and not related to prolonged survival (adjusted hazard ratio (95% CI): 1.07 (1.03–1.12)).

There are a limited number of population pharmacokinetics studies on CMS and colistin in patients on renal replacement therapy. Garonzik et al. [6] focused on CMS and colistin clearance in HD patients. The only study which focused on CMS and colistin clearance between two consecutive HD sessions were conducted by Jacob et al. [22], which reported that at the end of HD sessions, colistin concentrations drop to 1 to 1.5 mg/L, depending on the HD clearance used for stimulation. Karvanen et al. [21] and Garonzik et al. [6] reported a mean colistin concentration of 0.92–1.9 mg/L, which was below MIC (more than 2 mg/L) in critically ill patients receiving CVVHD with an adjusted dose of CMS. On the other hand, Leuppi-Taegtmeyer et al. [14] reported achieved concentration above MIC in all patients undergoing CVVHD, who received the standard dose of CMS (9 MIU loading dose followed by 3 MIU every 8 h). They also found that patients who weighed less than 60 kg receiving a maintenance dose of 2 MIU every 8 h did not achieve the target MIC concentration. Therefore, the study concluded that a loading dose of 9 MIU followed 8 h later by a maintenance dose of 3 MIU every 8 h independent of body weight is expected to achieve therapeutic colistin concentrations in patients undergoing CVVHD using low blood flow.

### 3.3. Pharmacokinetic Parameters of CMS and Colistin in Plasma at Steady State after CMS Administration

The pharmacokinetic parameters of CMS and colistin in critically ill patients’ plasma at a steady-state after CMS administration are summarized in Table 3, Table 4 and Table 5.

#### 3.3.1. Volume of Distribution (V_d_) of CMS and Colistin

The volume of distribution represents the apparent volume of a drug distributed based on the dose of the drug administered and describes the relationship between the dose and the resulting serum concentration [31]. Colistin methanesulfonate sodium is polyanion, and colistin is polycationic with a physiological pH of 7.4. Due to the large molecular weight, CMS and colistin cross the cellular membrane poorly [32]. In a healthy person, the V_d_ value of CMS and colistin has been consistent with the distribution restricted intravascularly [33]. However, wide inter-patient variability of V_d_ values of CMS and colistin was observed in pharmacokinetics studies of critically ill patients [3,6,9,23]. The V_d_ values of colistin ranged from 13.5 to 644 L. Most of the pharmacokinetics studies reported a V_d_ value of CMS and colistin less than 100 L [2,3,6,11,12,14,15,22]. Five studies reported a V_d_ value of colistin ranging from 100 to 250 L [4,9,10,20,23], and one study reported a higher V_d_ value of CMS and colistin, 915 ± 69.7 L and 644.5 ± 32.2 L, respectively [5]. Pharmacokinetics studies that involved patients with renal impairment and receiving renal replacement therapy reported low V_d_ values for CMS and colistin [6,14,21,22].

In critically ill patients, the V_d_ value is extremely dynamic as consequences of patients’ factors such as age, gender, severity or type of the disease, and albumin level may modify V_d_ value. Later, V_d_ value may be altered after intervention by the intensivist, such as fluid resuscitation, initiation of vasopressor agents, mechanical ventilator, extracorporeal therapy, and albumin administration [34,35,36]. An increase in V_d_ value may result in a decrease in the plasma antibiotic concentration, reduced CL, and likely an increased t_1/2_. Low plasma antibiotic concentration may affect the magnitude of the dose required. For example, Plachouras et al. [9] predicted a high V_d_ value for colistin (189 L) compared to Grégoire et al. [11], who predicted a low V_d_ value for colistin (25.7 L). The higher V_d_ value resulted in longer t_1/2_ values for colistin which needed a longer time to reach a steady state, as Plachouras et al. [9] observed. The V_d_ value of colistin in critically ill patients in the Grégoire et al. [11] study was consistent with the distribution in healthy volunteers [11,33], which did not represent the pathophysiological changes that occur in critically ill patients. Karaiskos et al. [3] also observed a lower V_d_ value (80.4 L) compared with earlier studies [4,6,10]. This was due to the volume of colistin being dependent on the available fraction of the A plus B form [3]. Using the estimated relative availability between the Karaiskos et al. [3] study and earlier studies [6,9,10] of 0.6 L, the 80 L estimated in the study would correspond to a volume of approximately 131 L in an earlier study, resulting in longer t_1/2_ values reported in the study.

#### 3.3.2. Clearance (CL) of CMS and Colistin

Colistin methanesulfonate sodium is predominantly cleared by renal excretion and the nonrenal clearance pathway. The nonrenal clearance pathway for CMS is hydrolysis in vivo to form a mixture of partially sulfomethylated derivatives converted to colistin and other pathways, such as hydrolysis of peptide bonds [32]. The renal clearance (CL_r_) of CMS was close to the glomerular filtration rate (GFR; around 120 mL/min) as reported by Garonzik et al. [6], and similar findings have been observed in studies of healthy volunteers [33]. In healthy volunteers’ plasma, the fraction of the CMS converted into colistin (f_m_) was estimated between 30 percent [33] and 60 percent [37]. In other studies, in healthy volunteers, approximately 62.5 percent of the CMS was excreted via urine within 24 h after dosing, whilst only 1.28 percent was present in the form of colistin [38]. Grégoire et al. [11] found a slightly different relationship in a formula that predicts renal clearance of CMS (CL_RCMS_) compared to the study by Garonzik et al. [6]. For example, in a CrCl value of 120 mL/min, Grégoire et al. [11] predicted a CL_RCMS_ value of 92 mL/min, whereas Garonzik et al. [6] predicted a CL_RCMS_ value of 123 mL/min, which is close to GFR.

The CL_r_ of colistin was very low due to its extensive tubular reabsorption after filtration at the glomerulus [25]. Colistin has a polycationic nature at physiological pH values and is unable to cross the cellular membrane efficiently. Thus, its tubular reabsorption is likely to involve one or more transport systems, such as organic cation transporters (OCTN1), peptide transporters (PEPT2), and megalin [39]. The renal reabsorption process is affected by the pH of the urine [32]. However, the exact mechanism of the elimination pathway of colistin remains unclear. The concentration of colistin in the blood that is excreted in urine is very low, but the urinary concentration of colistin after administration of CMS can be relatively high. This is due to the high concentration of CMS that is excreted within the urinary tract and converted into colistin [25].

The CL values of colistin reported ranging from 40–300 mL/min. There were discrepancies on reported CL values of colistin observed between studies. The discrepancies in the CL values of colistin were due to a difference in the estimated value of V_d_ between pharmacokinetics studies. As a result of a wide inter-patients’ variability, V_d_ values of colistin were observed in pharmacokinetics studies of critically ill patients. In patients with renal failure and receiving renal replacement therapy, the reported CL of colistin was between 13 to 137 mL/min [6,14,21,22]. The CRRT clearance accounted for 41 percent of total CMS clearance and 28 percent of total colistin clearance [14]. It was reported that the clearance of CMS and colistin via renal replacement therapy was related to the blood flow used in the renal replacement setting. In addition, higher blood flow used in renal replacement therapy increased the colistin clearance. The significant colistin clearance during CRRT was attributed to the absence of tubular reabsorption [14].

#### 3.3.3. Elimination Half-Life (t_1/2_) of CMS and Colistin

Following intravenous administration, the elimination t_1/2_ of antibiotic is the time it takes for the concentration of the antibiotic in the plasma or the total amount in the body to be reduced by 50 percent. Elimination of CMS was described by a two-compartment model in which the concentration declined biexponentially with a distribution half-life (t_1/2α_) and a terminal half-life (t_1/2β_). However, Grégoire et al. [11] did not observe a distribution phase, and a one-compartment model was used for CMS. Most of the pharmacokinetics studies reported CMS t_1/2_ values of 2 h [9,10,11,14,15], and other studies reported CMS t_1/2_ values of 3 to 5 h [2,3,6,21]. The elimination t_1/2_ of colistin was much longer than for CMS. The reported t_1/2_ value of colistin varies widely between 3 to 18 h. The reported colistin t_1/2_ value in critically ill patients was more prolonged than those observed in patients with cystic fibrosis [40] and in healthy volunteers [33].

Critically ill patients with sepsis experience dynamic changes of physiological conditions that alter the V_d_ and CL of antibiotics, which leads to discrepancies of t_1/2_ value between studies [35]. Colistin t_1/2_ is longer than that of many peptides [32,41]. The exact elimination pathway of colistin is still unknown, and the hydrolysis of colistin by enzymes, such as protease and peptides, has been proposed due to its peptide structure [32]. The cyclic structure of colistin protected it from proteolytic endopeptidase and the hydrophobic acyl chain in order to protect against exopeptidase. Therefore, the t_1/2_ value of colistin will be much longer than the value of another peptide [32,41]. The discrepancies are also likely due to a larger lag period between sample collection and sample analysis, causing falsely high colistin level concentrations at early points due to CMS hydrolysis to colistin, making the t_1/2_ value appear shorter than the true estimate [4,6,23].

### 3.4. Correlation between Pharmacokinetics Parameters of CMS and Colistin with Various Covariate Factors

Population pharmacokinetics models describe the relationship between dose, plasma concentration, and clinical covariates in particular patient populations [42]. Pharmacokinetics studies of CMS and colistin have identified a significant correlation between C_max_, V_d_, and weight [19,20]. Markou et al. [20] studied the steady-state pharmacokinetics of colistin without a loading dose and found significant negative correlations between C_max_ and V_d_ (*r* = −0.7). The V_d_ value was large, and the concentration of C_max_ was low. An increase in the V_d_ value of colistin as a result of a systemic inflammatory response syndrome (SIRS) related expansion of the extracellular fluid (ECF) volume and colistin may bind tightly to the membrane lipids of the cells in many body tissues, thereby increasing the V_d_ value and prolonging the t_1/2_ of the colistin [20,43]. Karnik et al. [12] examined the single-dose and steady-state pharmacokinetics of colistin receiving doses calculated on the basis of body weight and CrCl and found a significant correlation between C_max_ and body weight (*p* = 0.03). None of their subjects had CrCl less than 20 mL/min, and therefore the dosing was primarily based on body weight. Those patients whose body weight was higher than 60 kg had sufficient plasma colistin concentration and better survival than patients with a body weight less than 60 kg. Therefore, the study concluded that the dose recommendation should be based only on CrCl, and not body weight [12].

The suboptimal colistin concentrations during the first treatment day required the application of a loading dose. Population pharmacokinetic analysis by Garonzik et al. [6] found body weight as a relevant covariate affecting the V_d_ for CMS. Therefore, Garonzik et al. [6] suggested that a CMS loading dose should be calculated based on body weight. However, there were no significant correlations between clearance of either CMS or colistin against body weight, and thus they were not able to propose a weight-based dose for a maintenance dose of CMS [6]. A population pharmacokinetics analysis by Mohamed et al. [10] suggested that the application of a weight-based loading dose of CMS would have a limited impact on the initial rise in the concentration of formed colistin. Another study by Mohamed et al. [30] reported that loading dose followed by a maintenance dose every 8–12 h with an infusion duration of up to 2 h appears adequate for patients with normal or moderately impaired renal function.

Earlier pharmacokinetics studies have shown that no significant correlation was observed between colistin concentration and the CrCl value [9,10,20]. In contrast, Garonzik et al. [6] identified for the first time that renal function, expressed as CrCl, was an important covariate for the total clearance of both CMS and colistin. There was a significant correlation found between the concentration of CMS and colistin with CrCl [6]. In patients with preserved renal function, CMS is predominantly cleared by renal excretion, with only a small fraction of a dose converted to colistin. Thus, the total clearance of CMS is expected to decline with CrCl. In the case of decreased renal function, which resulted in reduced renal clearance of CMS, more CMS converted to colistin. Therefore, if CrCl is reduced, the concentration of colistin will increase due to the high conversion of CMS to colistin and vice versa. Kristoffersson et al. [2] has also observed a significant correlation between colistin concentration and CrCl.

Colistin has been reported to bind to α-1 acid glycoprotein (AGP). However, the actual binding of colistin to plasma component, albumin, AGP, lipoprotein, or globulin remain to be fully elucidated [44]. In critically ill patients, plasma protein binding of colistin was between 59–78 percent [10]. There was no obvious difference in plasma binding in critically ill patients and healthy volunteers [10]. Data regarding the correlation of pharmacokinetics parameters of CMS and colistin with various covariate factors in critically ill patients is limited and conflicted. Most of the pharmacokinetics studies involved a relatively small sample size, so the study was unable to conduct multivariate analysis to adequately identify all the predictors associated with outcome following CMS therapy [9,10,20,21]. A large number of statistical tests were performed despite the small sample size of pharmacokinetics studies giving rise to multiple statistical analysis problems [4].

## 4. Discussion

### 4.1. Heterogeneity in the Population Pharmacokinetics of CMS and Colistin in Critically Ill Patients

This review demonstrates vast inter-study discrepancies in pharmacokinetics parameter estimates. The population pharmacokinetic studies of colistin in critically ill patients are more variable than patients with cystic fibrosis and healthy volunteers [33,37,38,40]. A wide IIV of the pharmacokinetics parameter estimate has been observed in critically ill patients. The vast heterogeneity within and between studies in the critically ill patients’ population is the result of dynamic physiological changes due to the disease or intervention done by intensivists to correct the pathophysiological condition. Furthermore, patients in ICUs have more chronic comorbid illnesses, more severe acute physiologic derangements, are relatively more immunosuppressed, and are subject to increased selective pressure and increased colonization pressure, compared to patients in the general hospital population [45]. These physiologic changes can significantly affect the pharmacokinetics of CMS and colistin in this population. For example, endotoxins produced by Gram-negative bacteria in septic patients may trigger a systemic inflammatory response that affects the vascular endothelium, leading to increased capillary permeability and an increase in the V_d_. Alterations in the V_d_ value parameter may affect the clearance of antibiotics [35].

Differences in methodology and study protocol were observed in the pharmacokinetics studies. The discrepancies of the results are likely caused by heterogeneity in the study population characteristics, such as sample size, body weight, renal function, and the use of different CMS brands. A small number of patients were enrolled in the pharmacokinetics studies, including patients with preserved renal function and excluding patients with renal failure requiring renal replacement therapy; therefore, the results in the pharmacokinetics studies might not represent the full range of the critically ill patient population. In the Greek studies [3,9,10], the patients enrolled were of the median or mean body weight of 80 kg with preserved renal function and use of the same CMS brand, compared with another pharmacokinetics study which enrolled patients with variable body weight and renal function and used a different CMS brand [46]. The variability composition of the CMS formulation between different CMS brands used in the studies may have contributed to the interstudy discrepancies. The first study demonstrated that different brands of CMS from various countries had similar elemental compositions and comparable pharmacokinetics to CMS in rats and concluded that this led to different colistin concentrations [24]. Other than that, Karaiskos et al. [3] reported in their unpublished data in comparison of all six CMS brands that rates and extensions of colistin formation were slow for all six brands, including the brand Grégoire et al. [11] used.

The colistin composition in different brands may affect the CMS stability of preparation of CMS solution prior to administration. Colistin methanesulfonate sodium and colistin were shown to aggregate into micelles at a high concentration in an aqueous solution. Colistin methanesulfonate sodium’s critical concentration is 80,000 IU/mL, and rapid conversion of CMS to colistin occurs when the concentration is below the critical concentration (60 percent over 48 h) [47]. If the CMS were reconstituted sometime before its administration, the colistin would form in the solution resulting in a high concentration observed in the early sampling. The conversion of CMS to colistin can also occur when the duration of infusion is longer and vice versa. However, Karaiskos et al. [3] observed no difference in the pharmacokinetics of CMS and colistin between 30 min or 60 min infusions. The conversion of CMS in reconstituted solutions is of concern, particularly because active colistin is much more toxic than CMS [48].

Various dosing strategies were used between pharmacokinetics studies. A weight-based dose, fixed loading dose, dose based on CrCl, and the physician-based dose may lead to variability in the colistin levels achieved. The additionally administered inhalation of CMS may also augment the concentration values of colistin [4,6,15]. The differences in sampling strategies were observed between pharmacokinetic studies, in which most studies did the sampling after the steady-state was achieved. A study performed in India used a different methodology than other studies, which due to sparse sampling frequency, considered 4 and 8 h as the two time-points for calculating terminal elimination; this is generally not acceptable in modern pharmacokinetic studies [4]. Currently, there are no standard methods to measure CMS and colistin in plasma. Each pharmacokinetic study used its own developed method or did the same modification to the currently developed method to measure CMS and colistin levels. It is crucial to ensure the stability of CMS in the collected samples since CMS degradation, even at a low percentage, may affect the colistin concentration, especially when there is a high concentration of CMS and low colistin [9,49]. The ongoing conversion of CMS to colistin in vivo and in vitro cannot be prevented, but it can be minimized when the samples are collected and stored in a cool environment and analyzed after no more than 2 or 3 months of storage in order to avoid falsely high colistin concentration. The lag period of 2 to 10 months between sample collection and analysis may lead to a high steady-state concentration of colistin, as Ram et al. [4] observed.

### 4.2. Colistin Use in Critically Ill Patients

The pharmacokinetics of colistin in critically ill patients varies greatly. Colistin is administered intravenously as the non-active prodrug, colistin methanesulfonate sodium (CMS). Colistin methanesulfonate sodium is eliminated mainly by renal clearance. If CMS is cleared rapidly by the kidney, less CMS is converted to colistin resulted in low colistin that could lead to ineffective antibacterial therapy. In patients with normal renal function (1–2 MIU of CMS), approximately 30–60% of a dose of CMS is converted to colistin [1,2,3]. The renal clearance of CMS is much more efficient than the conversion of CMS to colistin [25]. Therefore, to achieve a targeted concentration of >2 mg/L, patients must receive four to five times the amount of CMS [3,9,10]. There was wide variability of colistin C_max_ values (0.6–8.7 mg/L) in plasma among single-dose studies receiving 2–3 MIU of CMS in critically ill patients with preserved renal function [9,12]. Almost none of the ICU patients achieved a colistin concentration of more than 2 mg/L in the Plachouras et al. [9] and Mohamed et al. [10] studies.

In the studies where the estimated t_1/2_ of colistin was high (14.4–18.5 h) reported that patients were exposed to suboptimal plasma colistin concentration for 2–3 days before reaching steady-state [6,9]. These prolonged t_1/2_ values take a longer time to achieve a steady state, which required three to five times the t_1/2_ values. Based on this consideration, the initiation of colistin therapy with a loading dose and a change in the dosing strategy for CMS has been suggested [3,9,10]. However, other studies challenged the rationale for a loading dose [4,5,11,12]. The differences in the methodology were observed, especially in studies performed in India, and discrepancies in the analytical methods could be an explanation [4,5,11,12].

The application of a loading dose of 6 MIU [10] and 9 MIU [3] of CMS have been studied. The colistin concentration was improved after a loading dose of 6 MIU but predicted PK/PD from the additional modeling suggested a higher dose is required to kill the wild-type *P. aeruginosa* strain [10]. Based on the findings and safety concerns, the study recommended a loading dose of 6–9 MIU in critically ill patients. After applying a loading dose of 9 MIU CMS, colistin C_max_ values were also highly variable (mean 2.66 ± 1.2 mg/L, and 0.9–5.1 mg/L range), and this approach was able to achieve the targeted colistin concentration in plasma within the first hours of the initiation [3]. The colistin concentration has been observed with values ranging from 0.68 to 8.72 mg/L at a steady state, and only 33 percent of the patients achieved colistin above 2 mg/L [3] but in a recent pharmacokinetic study [2] reported more than 90 percent of the patients had colistin concentration above 2 mg/L at a steady-state after receiving a loading dose of 9 MIU followed by 4.5 MIU every 12 h. The maintenance dose commenced 24 h after the administration of the CMS loading dose [3]. It is possible that the colistin concentration declined below the desired concentration by 12 h [30]. The maintenance dose should be commenced between 8–12 h after the administration of the CMS loading dose to achieve the desired concentration of colistin in the plasma of critically ill patients.

Critically ill patients will experience dynamic physiologic changes due to their pathological condition, and this phenomenon could enhance or reduce renal function. The amount of CMS dose required to achieve the desired concentration in critically ill patients is challenging. The apparent clearance of colistin depends on renal function [6]. A very wide interpatient variability in the apparent clearance of colistin at a given CrCl is observed [3,6,11]. Despite the appropriate loading dose being given to critically ill patients, it was difficult to achieve the targeted concentration of colistin in patients with moderate to good renal function [6]. Kristoffersson et al. [2] observed that 83 percent of patients with CrCl 80–119 mL/min and only 44 percent of patients with CrCl less than 120 mL/min achieved targeted concentration. The dosing algorithms for intravenous colistin in critically ill patients are proposed based on analysis of pharmacokinetics data from more than 200 critically ill patients with a wide range of renal functions [7]. Only 30–40% of patients with CrCl less than 90 mL/min are expected to achieve the desired concentration (>2 mg/L), even though they received the maximum dose of ~10.9 MIU CMS per day. Most of the patients (80 percent) may achieve a colistin concentration of less than 1 mg/L [7]. Therefore, research is needed to define optimal dosing strategies in patients with CrCl more than 80 mL/min. The targeted colistin concentration is more difficult to achieve in patients with augmented renal clearance (ARC). Augmented renal clearance is the enhanced elimination of the solute through the kidney with renal clearance above the expected baseline (above 130 mL/min). The development of ARC is related to SIRS and the hemodynamic manifestations of SIRS, including high cardiac output, due to this enhanced renal blood flow [50]. A dosage regimen higher than the current recommendation must be guided with TDM and AFC [16].

Medical interventions performed by the intensivist, such as the initiation of renal replacement therapy, may increase the clearance of CMS and colistin and lead to reduced CMS and colistin concentration in plasma. The colistin concentration that receives the same dose of CMS in critically ill patients requiring HD on days without an HD session is three times greater in critically ill patients having preserved renal function [22]. However, after the HD sessions, the colistin concentration dropped. Colistin methanesulfonate sodium and colistin can easily be filtered through dialysis membranes due to their molecular weight and the fact they are not bound in plasma. Therefore, a supplemental dose of 1.5 MIU should be administered after the HD session [8]. It was suggested that there was enhance elimination of colistin among patients receiving CRRT, compared to normal renal function [14]. Furthermore, colistin is known for its ability to be absorbed into different types of materials, including dialysis membranes, which could contribute to the colistin clearance mechanism [51]. Based on this consideration, patients receiving CRRT did not require a dosage, and the CMS dose should be used in the same manner and amount as in a patient with normal renal function [28]. However, as a consequence of the large IIV, TDM is advised for patients receiving CRRT [52].

Recently, more studies have been published to evaluate the pharmacokinetics of colistin in critically ill patients and correlate with clinical efficacy and renal function [2,4,5]. A prospective study in India evaluated the correlation of colistin pharmacokinetic with clinical efficacy and nephrotoxicity [5]. The study had a clinically favorable outcome without any significant nephrotoxicity in 50 percent of cases. Subtherapeutic colistin concentrations were observed in the “clinical failure” group [5]. Another pharmacokinetic study involved a large samples size reported, high colistin concentration was associated with poor renal function and was not related to prolonged survival [2]. Another study in India identified log-transformed colistin maximum concentration, area under the curve plasma concentration for 8 h, apparent total body clearance, and apparent volume of distribution was significantly associated with the safety outcome [4]. Understanding the fact that covariates may significantly influence variability in the pharmacokinetics of CMS and colistin helps to optimize the therapy and prevent toxicity. Although there were many covariates that were not clinically significant, combined effects should be taken into consideration.

## 5. Limitation

This review has limitations. Precise comparisons between studies were difficult because the published pharmacokinetic data derived from studies with different study designs, patient characteristics, and datasets were often reported as raw data, either mean or median for non-compartment analyses or derived from modeling analysis. Variability of the modeling method has the potential to introduce heterogeneity. There were studies reporting the pool of data from previous studies in their final modeling analyses. To enable comparisons of pharmacokinetics across studies, data were presented as the mean and standard deviation or predicted value from the pharmacokinetic modeling. Nevertheless, this review provides a comprehensive understanding of the CMS and colistin pharmacokinetics parameters in plasma after intravenous administration of CMS in critically ill patients.

## 6. Conclusions

The estimation of CMS and colistin pharmacokinetics parameters in critically ill patients revealed significant inter-study variations. The amount of colistin in the body is determined by the rate of CMS hydrolysis, CMS renal clearance, and clearance during dialysis therapy. The use of different CMS formulations may have contributed to the interstudy discrepancies. However, more research is needed to back up this assertion. The accomplishment of the appropriate colistin concentration in the plasma of critically ill patients could be achieved by administering a 9 MIU loading dose followed by a 4.5 MIU maintenance dose 8–12 h after the CMS loading dose for colistin with T_max_ 6–8 h. A loading dosage of 9 MIU followed by a maintenance dose of 3 MIU every 8 h is likely to achieve therapeutic colistin concentrations in renally compromised patients receiving renal replacement therapy. On non-HD days, 1.5 million international units (MIU) of CMS should be given twice daily to patients receiving HD. HD should be done at the end of a dosing period, and a 1.5 MIU additional dosage should be given following the HD session. More research is needed to determine the best dosing regimens for people who have a creatinine clearance of more than 80 mL/min. Because colistin has a narrow therapeutic index and high inter-individual variability, a dosing method that tailors colistin dosage based on blood colistin levels could help to strike a compromise between therapeutic efficacy and toxicity.

## Figures and Tables

**Figure 1 pharmaceuticals-14-00903-f001:**
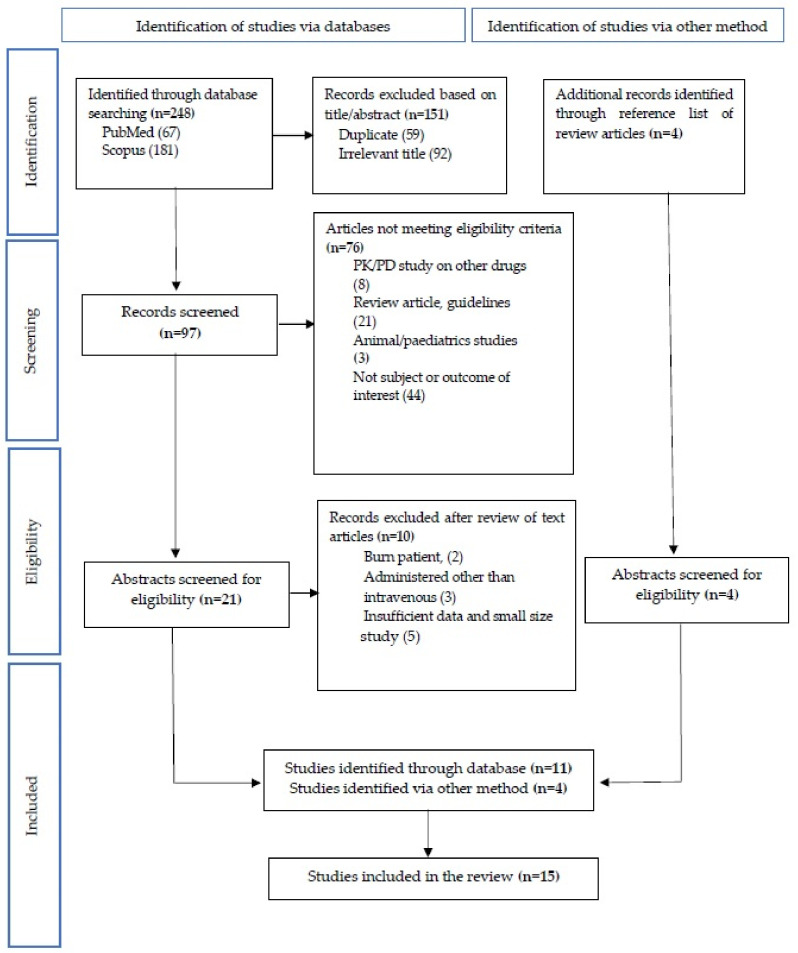
Studies identification and section.

**Table 1 pharmaceuticals-14-00903-t001:** Scope of literature review.

Criterion	Scope of Review
Population	Critically ill patients with or without renal replacement therapy
Intervention	Patient receiving intravenously administered colistin methanesulphonate sodium (CMS)
Outcome	Adult patients receiving single or multiple doses of colistin methanesulphonate sodium (CMS), the following outcomes were evaluated:
	C_max_, C_min_, C_ave_
	CL
	V_d_
	t_1/2_
Study design	Studies which samples were collected from subjects representative of target population with pharmacokinetics evidence able to inform any of the above-mentioned outcomes.

Abbreviations: C_max_: Maximum concentration; C_min_: Minimum concentration; C_ave_: Average concentration; CL: Clearance; V_d_: Volume of distribution; t_1/2_: Half-life.

**Table 2 pharmaceuticals-14-00903-t002:** Comparison of CMS and colistin concentration in plasma after administration of a first dose of CMS.

Author year	Brand	First Dose	Reported Value	CMS		Colistin	
				T_max_ (h)	C_max_ (mg/L)	T_max_ (h)	C_max_ (mg/L)
Plachouras et al. 2009 [9]	Colistin	3 MIU	Predicted value	NR	~3.5	7	0.6
Karaiskos et al. 2015 [3]	Colistin	9 MIU	Predicted value	1	17	7	2.3
Grégoire et al. 2014 [11]	Colimycine	2 MIU	Predicted value	1	6.5	3	2
Jacob et al. 2015 [22]	Colimycine	1–9 MIU	Predicted value	1	9	6	3
Mohamed et al. 2012 [10]	Colistin	6 MIU	Mean ± SD	NR	NR	8	1.4 ± 0.7
Karvanen et al. 2012 [21]	Colistin	2 MIU	Mean ± SD	0.6 ± 0.1	6.9 ± 2	6	0.3 ± 0.1
Kristoffersson et al. 2020 [2]	Colimycine	9 MIU	Mean ± SD	0.75	33.3 ± 9.3	0.75	0.8 ± 0.4
Karnik et al. 2013 [12]	Xylistin^TM^	2 MIU	Mean ± SD	NR	NR	1	7.8 ± 5
Moni et al. 2020 [5]	Coly-monas^®^	9 MIU	Mean ± SD	0.5 ± 0.2	14.5 ± 136	2.75 ± 1.8	2.66 ± 1.2

C_max_: maximum concentration; C_min_: minimum concentration; C_ave_: Average concentration; CL: Clearance; CMS: Colistin methanesulfonate sodium; h: hour; L: Litre mg/L: miligram/litre; ml/min: mililitre/minute; MIU: million units; NR: Not reported; t_1/2_: Half-life; V_d_: Volume of distribution.

**Table 3 pharmaceuticals-14-00903-t003:** Comparison CMS pharmacokinetics parameters in plasma at steady state.

Author Year	Dose	Reported Value	Plasma Concentration			Pharmacokinetic Parameters		
			C_max_ (mg/L)	C_min_ (mg/L)	C_ave_ (mg/L)	t_1/2_ (h)	CL (ml/min)	V_d_ (L)
Plachouras et al. 2009 [9]	3 MIU 8 hourly	Predicted value	8 to 9	NR	NR	0.05, 2.3	228.3	13.5, 28.9
Mohamed et al. 2012 [10]	6 MIU then 1–3 MIU 8 hourly	Predicted value	NR	NR	NR	0.03, 2.2	218.3	11.8, 28.4
Matthieu et al. 2014 [15]	2 MIU 8 hourly	Predicted value	NR	NR	NR	2.7	64.6	15.3
Grégoire et al. 2014 [11]	6 MIU/day 8–12 hourly	Predicted value	6.5	NR	NR	1.9	110.1	18.2
Karaiskos et al. 2015 [3]	9 MIU then 4.5 MIU 12 hourly	Predicted value	NR	NR	NR	5.4	90	28
Kristoffersson et al. 2020 [2]	9 MIU then 4.5 MIU 12 hourly	Predicted value	NR	NR	NR	5.6	27	13
Markou et al. 2008 [20]	3 MIU 8–12 hourly	Mean ± SD	NR	NR	NR	2.9 ± 1.2	227.7 ± 96.7	NR
Karvanen et al. 2012 [21]	2 MIU 8 hourly	Mean ± SD	6.9 ± 2.8	1.5 ± 0.6	NR	3.3	137.2 ± 51.2	NR
Leuppi-Taegtmeyer et al. 2019 [14]	6–9 MIU then 3 MIU 8 hourly	Mean ± SD	NR	1.3 ± 0.7	5.0 ± 1.9	2.1 ± 0.5	70.6 ± 28.9	12.9 ± 5.8
Moni et al., 2020 [5]	9 MIU then 3 MIU 8 hourly	Mean ± SD	3.8 ± 2.2	0.3 ± 0.3	1.7 ± 1.0	NR	NR	915 ± 69.7

C_max_: maximum concentration; C_min_: minimum concentration; C_ave_: Average concentration; CL: Clearance; CMS: Colistin methanesulfonate sodium; h: hour; L: Litre mg/L: miligram/litre; ml/min: mililitre/minute; MIU: million units; NR: Not reported; t_1/2_: Half-life; V_d_: Volume of distribution.

**Table 4 pharmaceuticals-14-00903-t004:** Comparison of colistin pharmacokinetics parameters in plasma at steady state.

Author Year	Dose	Reported Value	Plasma Concentration			Pharmacokinetic Parameters		
			C_max_ (mg/L)	C_min_ (mg/L)	C_ave_ (mg/L)	t_1/2_ (h)	CL (ml/min)	V_d_ (L)
Plachouras et al. 2009 [9]	3 MIU 8 hourly	Predicted value	2.3	NR	NR	14.4	151.5	189
Garonzik et al. 2011 [6]	1–5 MIU 8–24 hourly	Predicted value	NR	NR	2.4	4.6	45.3	45.1
Mohamed et al. 2012 [10]	6 MIU then 1–3 MIU 8 hourly	Predicted value	NR	NR	<2	18.5	136.7	218
Grégoire et al. 2014 [11]	6 MIU/day 8–12 hourly	Predicted value	2	NR	NR	3.2	94.3	25.7
Matthieu et al. 2014 [15]	2 MIU 8 hourly	Predicted value	4.7	0.15	NR	4.3	53.1	13.7
Jacob et al. 2015 [22]	1–9 MIU then 1.5 MIU 12 hourly	Predicted value	3.6	NR	NR	9.8	33.3	28.3
Karaiskos et al. 2015 [3]	9 MIU then 4.5 MIU 12 hourly	Predicted value	0.68–8.72	NR	NR	11.2	81.7	80.4
Kristoffersson et al. 2020 [2]	9 MIU then 4.5 MIU 12 hourly	Predicted value	NR	NR	>2	12, 17 and 25 *	50.5	81.2
Markou et al. 2008 [20]	3 MIU 8–12 hourly	Mean ± SD	2.9 ± 1.2	1.0 ± 0.4	NR	7.4 ± 1.7	226.7 ± 96.7	139.9 ± 60.3
Imberti et al. 2010 [23]	2 MIU 8 hourly	Mean ± SD	2.21 ± 1.1	1.03 ± 0.7	NR	5.9 ± 2.6	346 ± 240	120 ± 88
Karvanen et al. 2012 [21]	2 MIU 8 hourly	Mean ± SD	NR	NR	0.9 ± 0.5	NR	315.2 ± 99.2	NR
Karnik et al. 2013 [12]	2 MIU 8–12 hourly	Mean ± SD	8.6 ± 5.7	0.8 ± 0.5	2 ± 0.8	8.6 ± 5.7	74.9 ± 16.6	40.5 ± 21.2
Leuppi-Taegtmeyer et al. 2019 [14]	6–9 MIU then 3 MIU 8 hourly	Mean ± SD	NR	3.9 ± 0.8	4.7 ± 1.5	17.8 ± 7.5	50.9 ± 15.6	72.2 ± 24.6
Moni et al. 2020 [5]	9 MIU then 3 MIU 8 hourly	Mean ± SD	2.4 ± 2.2	1.5 ± 0.9	2 ± 1.2	NR	NR	644.5 ± 32.2
Ram, et al. 2020 [4]	2 MIU 8 hourly	Mean ± SD	10.6 ± 1.6	0.7 ± 0.2	NR	2.9 ± 0.43	130 ± 40	33 ± 6.6

C_max_: maximum concentration; C_min_: minimum concentration; C_ave_: Average concentration; CL: Clearance; CMS: Colistin methanesulfonate sodium; h: hour; L: Litre mg/L: miligram/litre; ml/min: mililitre/minute; MIU: million units; NR: Not reported; t_1/2_: Half-life; V_d_: Volume of distribution. *Patients with creatinine Clearance of 120 mL/min, 80 mL/min and 50 mL/min.

**Table 5 pharmaceuticals-14-00903-t005:** Comparison of CMS and colistin pharmacokinetics parameters in plasma in patients receiving renal replacement therapy.

Author Year	Dose	Reported Value	CMS				Colistin			
			t_1/2_ (h)	CL (ml/min)		V_d_ (L)	t_1/2_ (h)	CL (ml/min)		V_d_ (L)
				Total	Dialysis			Total	Dialysis	
Garonzik et al. 2011 [6]	1–5 MIU 8–24 hourly	Predicted value	4.6	103.3	94.8 (HD) 64.2 (CRRT)	18.7	4.6	45.3	56.7 (HD) 34.3 (CRRT)	45.1
Jacob et al. 2015 [22]	1–9 MIU then 1.5 MIU 12 hourly	Predicted value	2.1	113	90 (HD)	21	9.8	33.3	137 (HD)	28.3
Karvanen et al. 2012 [21]	2 MIU 8 hourly	Mean ± SD	3.3	137.2 ± 51.2	32.3 ± 13.3 (CRRT)	NR	NR	315.2 ± 99.2	71.7 ± 21.7 (CRRT)	NR
Leuppi-Taegtmeyer et al. 2019 [14]	6–9 MIU then 3 MIU 8 hourly	Mean ± SD	2.1 ± 0.5	70.6 ± 28.9	26.3 ± 3.7 (CRRT)	12.9 ± 5.8	17.8 ± 7.5	50.9 ± 15.6	13.3 ± 3.2 (CRRT)	72.2 ± 24.6

CMS: Colistin methanesulfonate sodium; CL: Clearance; CRRT: Continuous Renal Replacement Therapy; HD: Hemodialysis; h: hour; L: Liter; LD: Loading dose; mg/L: Milligram/liter; ml/min: Milliliter/minute; MIU: million units; NR: Not reported; PK: Pharmacokinetics; SD: Standard deviation t_1/2_: Elimination half-life; V_d_: Volume of distribution.

## Data Availability

Not applicable.

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
