# Peer review of "Population Pharmacokinetics of Colistin Methanesulfonate Sodium and Colistin in Critically Ill Patients: A Systematic Review"

_pharmaceuticals, 2021, doi:10.3390/ph14090903_

Round 1

Reviewer 1 Report

Dear Editor,

The paper intention was to compare the pharmacokinetic parameters of Colistin methanesulfonate sodium and Colistin to optimize the dosage regimen in critically ill patients.

Introduction should be more updated, with a difference between iv administration and the inhalatory administration. E.g., Colistin administered inhalatory is not properly cited (More recent papers should be mentioned: Choe J, Sohn YM, Jeong SH, et al. Inhalation with intravenous loading dose of colistin in critically ill patients with pneumonia caused by carbapenem-resistant gram-negative bacteria. Ther Adv Respir Dis. 2019;13:1753466619885529. doi:10.1177/1753466619885529). Why the authors did not make references to studies previous 2020?

Materials and method should be improved. The PICO criteria are not very clear. I wonder why burn patients were not included (as seen in Eligibility - figure 1).

Results. The brand is important to explain the differences for Cmax and T max between studies and this mention in the tables might improve the chapter 3.2.

Discussions. The authors underline well the limitations of the paper.

Author Response

Thank you

Reviewer 2 Report

The manuscript by Zabidi et al. aims to standardize colistin treatment regimens by analyzing literature data and presenting lengthy tables with them. As attractive as the idea may be to physicians, the manuscript contains tables with data and no conclusions, except the last part. It is challenging to comprehend an avalanche of literature reports and associated details. At this stage, it is suitable for supplementary material.

The manuscript requires better organization and a clear presentation. Practically, every table could be replaced by a figure with core data in the Supplementary Information section.

Author Response

Thank you

Reviewer 3 Report

Dear Editor,

The paper intention was to compare the pharmacokinetic parameters of Colistin methanesulfonate sodium and Colistin to optimize the dosage regimen in critically ill patients.

Introduction should be more updated, with a difference between iv administration and the inhalatory administration. E.g., Colistin administered inhalatory is not properly cited (More recent papers should be mentioned: Choe J, Sohn YM, Jeong SH, et al. Inhalation with intravenous loading dose of colistin in critically ill patients with pneumonia caused by carbapenem-resistant gram-negative bacteria. Ther Adv Respir Dis. 2019;13:1753466619885529. doi:10.1177/1753466619885529). Why the authors did not make references to studies previous 2020?

Materials and method should be improved. The PICO criteria are not very clear. I wonder why burned patients were not included (as seen in Eligibility - figure 1).

Results. The brand is important to explain the differences for Cmax and T max between studies and this mention in the tables might improve the chapter 3.2.

Discussions. The authors underline well the limitations of the paper.

The manuscript should be more systematized.

Author Response

Thank you

Reviewer 4 Report

The manuscript entitled "Population Pharmacokinetics of Colistin Methanesulfonate Sodium and Colistin in Critically Ill Patients: A Systematic Review" is fascinating.

 This manuscript deals with the comprehensive understanding of CMS and colistin pharmacokinetics parameters.

 It was very interesting to read the complete manuscript. After reading and thoroughly analyzing the data (text and tables) presented by the authors, I have decided to recommend a significant revision of the manuscript. I have tried to mention some of the crucial points to improve the quality of the paper for future submission: 
 The introduction should be improved. It should include more recent trends, currently known research findings. 

 Present your novelty or impact on the work/field in the introduction section concerning current findings. Authors must provide the significance and originality of this study in more detail.
Tables need a clear description. 

 The conclusion should be elaborated based on the results presented.
 The manuscript demands thorough grammatical English language correction.
 In conclusion, in my opinion, the work could be accepted after major revision for publication in Pharmaceuticals.

Author Response

Thank you

Round 2

Reviewer 1 Report

Dear Authors,

The article is improved.

However, there is an important non-concordance between the numbers of articles in the abstract/results sections and Fig. 1. Moreover, adding the numbers from Fig 1, there are other results: 248 + 67 + 181 - 151 - 59 - 92  is not 97. Something is missing.

All the references should be with "[ ]" (Karvanen et al. 2012 (9) from Table 2 should be corrected with [9])

Table 3 and 4: Cmax, Cmin, Cave are considered also Pharmacokinetic parameters, therefore, for Tmax, ... should be Other Pharmacokinetic parameters.

Author Response

Thank you.

Reviewer 2 Report

The improved version of the manuscript is much better. The data is presented clearly, extensive tables have been replaced by shortened versions.

I have found only one error in Table 2. The reference to Karvanen et al. should be listed as [21] instead of (9).

The Conclusions section recommends the first dose of 9MIU to be followed by 4.5MIU after 8h. However, Table 2 shows that Tmax for selected brands can be as short as 0.75h and as long as 8h. The recommendation does not mention different PK/PD properties of brands despite having it discussed in the text extensively. Is giving such a recommendation justified when the values may depend on the brand used? It would be helpful to clarify the issue.

Author Response

Point 1: I have found only one error in Table 2. The reference to Karvanen et al. should be listed as [21] instead of (9).

Response 1: Thank you for your observation and pointing out the mistake made; we've double-checked the references list, and modifications to Karvanen et al., 2012 [21] references have been indicated in red text.

Point 2: The Conclusions section recommends the first dose of 9MIU to be followed by 4.5MIU after 8h. However, Table 2 shows that Tmax for selected brands can be as short as 0.75h and as long as 8h. The recommendation does not mention different PK/PD properties of brands despite having it discussed in the text extensively. Is giving such a recommendation justified when the values may depend on the brand used? It would be helpful to clarify the issue.

Response 2: Thank you for your invaluable query. We have modified the statement as below to clarify the issue (Page 19, line 736-740).

The use of different CMS formulations may have contributed the interstudy discrepancies. However, more research is needed to back up this assertion. The accomplishment of the appropriate colistin concentration in the plasma of critically ill patients could be achieved by administering a 9MIU loading dose followed by a 4.5MIU maintenance dose 8-12 hours after the CMS loading dose for colistin with Tmax 6-8 hours.

Reviewer 4 Report

The authors addressed the comments adequately, and the manuscript improved well. 

Author Response

Point 1: The authors addressed the comments adequately, and the manuscript improved well. 

Response 1: Thank you for invaluable comments.